# Surgically Induced Menopause—A Practical Review of Literature

**DOI:** 10.3390/medicina55080482

**Published:** 2019-08-14

**Authors:** Cristina Secoșan, Oana Balint, Laurențiu Pirtea, Dorin Grigoraș, Ligia Bălulescu, Răzvan Ilina

**Affiliations:** 1Department of Obstetrics and Gynecology, University of Medicine and Pharmacy “Victor Babeş”, 300041 Timişoara, Romania; 2Department of Obstetrics and Gynecology, County Hospital Timişoara, 300172 Timişoara, Romania; 3Department of Surgery, University of Medicine and Pharmacy “Victor Babeş”, 300041 Timişoara, Romania

**Keywords:** menopause, surgery, complications, oophorectomy, salpingectomy

## Abstract

Menopause can occur spontaneously (natural menopause) or it can be surgically induced by oophorectomy. The symptoms and complications related to menopause differ from one patient to another. We aimed to review the similarities and differences between natural and surgically induced menopause by analyzing the available data in literature regarding surgically induced menopause and the current guidelines and recommendations, the advantages of bilateral salpingo-oophorectomy in low and high risk patients, the effects of surgically induced menopause and to analyze the factors involved in decision making.

## 1. Introduction

Menopause represents the permanent cessation of menstrual periods and the loss of fertility due to the loss of ovarian function. It can occur spontaneously (natural menopause) or it can be surgically induced by bilateral oophorectomy.

The symptoms and menopause-related complications are caused by decreased estrogen levels. They are numerous and their severity is variable from one patient to another: hot flushes and night sweats, psychological changes (such as depression, and impaired concentration), insomnia, vaginal dryness, skin changes (such as thinning and decreased elasticity) [1,2]. Sexual function is also affected. The risk of sexual dysfunction in menopausal women increases with the drop-in estrogen levels and the aging process [3].

This article represents a review of the currently available data regarding the major controversies related to surgically induced menopause, such as: the effects of surgically induced menopause on women’s health, the current guidelines and recommendations for ovarian preservation versus oophorectomy in oncologic surgery and salpingo-oophorectomy (SO) in low and high-risk patients, the advantages of bilateral salpingo-oophorectomy, the residual ovary syndrome, the effects of surgically induced menopause and also the factors associated with undergoing bilateral SO at the time of hysterectomy for benign conditions.

## 2. Effects on Women’s Health

The abrupt discontinuation of ovarian function (by oophorectomy) in pre-menopausal women is associated with more severe consequences than natural menopause, such as increased overall mortality rate (16.8% versus 13.3% in patients with ovarian conservation), and increased rates of pulmonary and colorectal cancer, coronary disease, stroke, cognitive impairment, Parkinson’s disease, psychiatric disorders, osteoporosis and sexual dysfunction [4,5,6,7].

Although for many years it has been thought that the ovary retains a degree of androgen production even after menopause, more recent immunocytochemical evidence shows that the postmenopausal ovary lacks steroidogenic enzymes and luteinizing hormone (LH)receptors and the major source of postmenopausal circulating androgens being in fact the adrenal gland [8].

## 3. Ovarian Preservation or Oophorectomy in Oncologic Surgery—Recommendations for Cervical, Endometrial and Ovarian Cancer

Ovarian preservation in oncologic surgery depends on the stage of the disease and the patient’s age. The European Society for Medical Oncology, European Society of Gynaecological Oncology and European Society for Radiotherapy and Oncology (ESMO-ESGO-ESTRO) cervical cancer guidelines (2018), for the management of stage T1b1/T2a1 (clinically visible lesion of 4 cm or less) recommend ovarian preservation in premenopausal patients with cellular squamous cell carcinoma or adenocarcinoma (Human Papilloma Virus associated) and bilateral salpingectomy should be considered [9].

The ESMO-ESGO-ESTRO consensus on endometrial cancer recommended that standard surgery for endometrial cancer is total hysterectomy with bilateral salpingo-oophorectomy. Ovarian preservation may be considered only in patients younger than 45 years with grade 1 EEC (early endometrial cancer) with <50% myometrial invasion and no apparent ovarian or extra-uterine pathology. Ovarian preservation is not recommended for patients with a family history of cancer, including the risk of ovarian cancer. Genetic tests are recommended [10].

Regarding ovarian cancer, the ESMO-ESGO consensus recommendations are that fertility sparing surgery (bilateral salpingo-oophorectomy and complete surgical staging) can be performed at all stage IA and IC1 patients, for low risk ovarian carcinomas. Ovarian preservation for invasive epithelial ovarian cancers is not recommended at a higher stage than FIGO I (The International Federation of Gynecology and Obstetrics) [11].

Regarding borderline ovarian tumors, the choice of radical intervention versus ovarian preservation is still controversial. Approximately 1/3 of ovarian borderline tumors occur in women under 40 years, many desiring preservation of at least one ovary in order to maintain fertility and to avoid the symptoms of premature menopause [12,13]. In stage I, unilateral salpingo-oophorectomy or cystectomy may be an option for unilateral tumors. The recurrence rate after unilateral salpingo-oophorectomy is considered to be relatively low (7%), but much higher after cystectomy (23%) [14].

## 4. Indications for Salpingo-Oophorectomy in Low and High-Risk Patients—Current Guidelines and Recommendations

Elective salpingo-oophorectomy represents the removal of ovaries and fallopian tubes in a woman without indication for this procedure. Risk-reducing salpingo-oophorectomy is defined as the removal of ovaries and fallopian tubes in a woman with hereditary ovarian cancer syndrome.

The guideline for salpingo-oophorectomy was presented in the American College of Obstetricians and Gynecologists (ACOG) Practice Bulletin in January 2008. It states that bilateral salpingo-oophorectomy should be offered to women with BRCA1 and BRCA2 mutation after finishing their reproductive life. Women with a family history suggestive for BRCA1 and BRCA2 mutation require genetic evaluation and counseling. For women at high risk of ovarian cancer, risk-reducing salpingo-oophorectomy should include careful peritoneal cavity inspection, peritoneal lavage, complete salpingectomy and proximal ligation of ovarian vessels. Ovarian preservation in premenopausal women without increased genetic risk for ovarian cancer is recommended. Because of the risk of ovarian cancer in postmenopausal women, oophorectomy should be considered at the time of hysterectomy. Patients with endometriosis, pelvic inflammatory disease or chronic pelvic pain have an increased risk for surgical reintervention and the risk of a subsequent intervention for ovarian pathology should be weighed against the benefits of ovarian preservation [15].

Indications for salpingo-oophorectomy in low-risk patients:benign ovarian tumors—in cases where cystectomy, enucleation or partial oophorectomy are not feasible;tubal-ovarian abscess without response to antibiotic treatment;adnexal torsion complicated by necrosis;endometriosis [16].

Indications for salpingo-oophorectomy in high-risk patients:patients with gynecological malignancies or ovarian metastatic cancer—for staging and treatment;patients with inherited genetic mutations.

In 2003, Antoniou associated the presence of BRCA1 mutation with a 39% risk for developing ovarian cancer. For BRCA 2 mutation the risk of ovarian cancer is 11–17% [17]. The American Society of Clinical Oncology associated Lynch syndrome with a 9–12% risk for developing ovarian cancer [16].

## 5. Advantages of Bilateral Salpingo-Oophorectomy (SO)

### 5.1. Prevention of Ovarian Cancer

In low risk patients, according to a study by Jacoby et al. 2014, the global risk for developing ovarian cancer in low risk patients was 1.4% and was influenced by numerous factors. In low risk patients, performing SO is considered to completely eliminate the risk of ovarian cancer development [18]. However, there have been reports in the literature of cases of epithelial ovarian cancer after bilateral salpingo-oophorectomy or primary peritoneal carcinomatosis after bilateral salpingo-oophorectomy [19,20].

In high risk patients, risk reducing bilateral SO is reported to reduce by 70–80% the risk of ovarian cancer, mortality by cancer and mortality in general [21].

The Society of Gynecologic Oncology issued the recommendations for the prevention of ovarian cancer in high risk patients [22]. Risk-reducing bilateral salpingo-oophorectomy is recommended between 35–40 years to reduce risk in women with high genetic risk of ovarian cancer. The age for bilateral SO can be individualized according to the age of cancer onset within the family or personal desires. This topic is still debated because in the context of a patient with Hereditary Ovarian Cancer Syndrome we are discussing a personalized surveillance program, complete genetic evaluation of all 16 genes involved in the ovarian cancer genesis, application of chemoprevention therapy, and in case of the presence of cumulative risk factors, performing bilateral SO is necessary. Salpingectomy can be considered after completion of the reproductive life in women at a high risk of ovarian cancer who do not want SO. Salpingectomy may be considered in moderate-risk women who undergo hysterectomy, other pelvic surgery or tubal ligation.

Regarding prophylactic salpingectomy, meaningful conclusions are not yet available. It was introduced in 2010 following evidence of the origin of epithelial ovarian cancer with a starting point in the fallopian tube. Numerous associations have positive opinions regarding the procedure (Royal Australian and New Zealand College of Obstetricians and Gynecologists; Society of Gynecologic Oncology; Royal College of Obstetricians and Gynecologists; American College of Obstetricians and Gynecologists; Commission Ovary of the Arbeitsgemeinschaft Gynalologische Onkologie). On the other hand, some authors have reported possible disadvantages of the procedure, such as alteration of the ovarian reserve (results are contradictory), prolonged operative time and technical difficulties in the case of vaginal hysterectomy or caesarean section [23]. 

Another possible risk reducing strategy for high-risk women who are not ready to undergo menopause and its consequences is represented by bilateral salpingectomy with ovarian retention. This strategy was proposed and discussed at the 24th Annual Ella T. Grasso Ovarian Cancer Symposium in 2008 as a temporary intermediate step for a later bilateral oophorectomy. Its advantages and disadvantages are presented in Table 1.

### 5.2. Prevention of Breast Cancer

In low risk patients the results are contradictory. In 2009, Parker reported a decreased incidence of breast cancer (0.75%), but a similar mortality rate [24]. In 2011, Jacoby reported a decreased incidence only in patients under the age of 40 at the time of the BSO, without hormonal substitution [25].

In high risk patients, the risk for breast cancer was reduced by 37–54% [17]. Terry et al. performed a prospective cohort study including 17.917 breast cancer patients (7.2% positive for BRCA1 and BRCA2 gene mutations) and concluded that prophylactic SO was not considered effective in reducing the risk of breast cancer [26].

### 5.3. Decreasing the Risk of Reintervention after Hysterectomy

The risk of reintervention after hysterectomy in a patient with adnexal preservation after hysterectomy is estimated between 0.89–5.5% [27,28]. Pain (71%) and the presence of an adnexal mass (25%) are considered the most common causes of reintervention [29]. Casiano et al., compared the percentage of reinterventions in 5000 patients with bilateral SO and 5000 patients with adnexal preservation at the time of hysterectomy and performed a 30 year follow up. They found that the incidence of reinterventions was higher in the group with SO, but the difference was not significant [30].

## 6. The Residual Ovary Syndrome

The term “residual ovary syndrome” describes the pelvic symptomatology subsequent to hysterectomy with ovarian preservation. The incidence of this syndrome and the need for subsequent reintervention were studied by Dekel et al. in a retrospective study including 2561 patients with hysterectomy (with or without oophorectomy) over a period of 20 years. Their results showed that residual ovarian syndrome affects 1/35 patients with hysterectomy. The main indications for reintervention were chronic pelvic pain (71.3%) or an asymptomatic pelvic mass diagnosed during routine examination (24.6%). They concluded that routine oophorectomy was justified for premenopausal women over 45 years [29].

## 7. The Effects of Surgically Induced Menopause

The effects of surgically induced menopause are numerous, global survival, the cardiovascular system, cognitive function, sexual function and bone loss being the main affected, as shown in Table 2 and detailed below.

### 7.1. On Global Survival

Surgically induced menopause has been linked to an increased mortality risk, according to several authors. Parker et al. aimed to identify the optimal strategy to prolong the survival of non-high-risk patients for ovarian cancer and analyzed the survival rate on a period of up to 80 years, in patients with or without hysterectomy and with or without oophorectomy. They concluded that conservation of ovaries in women under 65 years with low risk for ovarian cancer provided long-term survival benefit. Hysterectomy is significant only in women with oophorectomy and less important in the case of a patient with conservation of ovaries [31].

Also, Ossewaard et al. showed a decrease by 2% of total mortality with each year of menopause delay [32]. Rocca et al. found that the mortality risk increased in patients with bilateral oophorectomy under 45 years of age [33]. In 2009 and then in 2013, Parker confirmed these results, demonstrating that oophorectomy was associated with an increased risk of death regardless of cause for women under 50 years who were not treated with hormone replacement therapy (HRT) [24,34]. The same conclusion was reported by other recent studies. In 2014, Gierach observed that women with bilateral SO under 35 years of age had an increased risk of death regardless of the cause, but the risk became similar for women over 50 years of age [35]. Additionally, Mytton showed that ovarian conservation was associated with lower death rates regardless of the cause [36].

### 7.2. On Cardiovascular Disease

Early menopause and surgically induced menopause are associated with an increased risk of cardiovascular disease. The proposed mechanism is represented by accelerated atherosclerosis due to the increased levels of atherogenic lipoproteins secondary to a hypoestrogenic state. Studies show an increased risk for cardiovascular diseases in women with oophorectomy without HRT, while following a hormone replacement therapy eliminates the risk [37]. Parker et al. found an increased risk for cardiovascular diseases regardless of the age of women with oophorectomy. The risk increases in women <45 years, with a maximum risk for women <50 years without HRT [24]. Rivera et al. stated that the highest risk for cardiovascular diseases was observed in women with oophorectomy <45 years without HRT or interrupted HRT [38]. Additionally, Ingelsson et al. demonstrated that hysterectomy with bilateral oophorectomy in women <50 years increases the risk for cardiovascular diseases by 40% [39].

### 7.3. On Cognitive Function

Large cohort studies have analyzed the impact of surgically induced menopause on cognitive function. The Mayo Clinic Cohort of Oophorectomy and Aging (SUA) analyzed 2390 women who were followed for a 25–30 year period. They found that women with bilateral/unilateral oophorectomy before menopause had an increased risk of cognitive impairment, dementia or Parkinsonism. The risk increased with a younger age at the time of oophorectomy. The association was similar regardless of the indication for oophorectomy [40].

The Religious Orders Study and Memory and Aging Project (SUA) followed 1884 from two separate cohorts over a period of 18 years. They reported that younger age at the time of SO was associated with a rapid decline in global cognitive function and a high risk of Alzheimer’s disease. The use of hormonal substitution therapy for at least 10 years, with initiation in the first 5 years after menopause onset, was associated with improvement in cognitive decline. These associations were not observed in women with natural menopause [41].

### 7.4. On Sexual Function

Studies report negative consequences on sexual function in women with oophorectomy without hormone replacement therapy, such as: decreased libido and difficult sexual arousal, a 3-fold risk of anorgasmia; and increased risk of hypoactive sexual desire disorder [42,43]. A slower decline in global sexual function (frequency of sexual activities, dyspareunia, libido) was found in women with hysterectomy without SO or with unilateral SO, than in women with bilateral SO [43].

### 7.5. On Bone Loss

There are several studies that describe the effects of menopause on bone loss. More than 75% of the bone mass lost in the first 20 years after menopause is attributed to estrogen deficiency and not to the aging process [44]. Regarding surgically induced menopause, bilateral oophorectomy in women under 45 years of age is considered a risk factor for osteoporosis [45]. Bone loss reaches up to 20% in the first 18 months after bilateral oophorectomy [46]. Moreover, bilateral oophorectomy increases the risk of osteoporosis even in postmenopausal women compared to those with intact ovaries [45,46,47]. However, Karim showed in a study published in the USA in 2011 that loss of bone mass was similar regardless of the type of menopause and hormonal substitution therapy had a protective effect limited to the usage duration, the protective effect disappearing two years after the treatment was stopped [48].

## 8. Factors Associated with Undergoing Bilateral Salpingo-Oophorectomy at the Time of Hysterectomy for Benign Conditions

Bilateral salpingo-oophorectomy during hysterectomy for benign pathology is a common procedure in premenopausal women performed in 40% of cases for women between 40–44 years old and in 63% of cases for women between 45–49 years old [49].

Several studies analyze the factors influencing the decision of performing a salpingo-oophorectomy, taking into account, first of all, the physician-patient relationship.

The Nationwide Inpatient Sample database failed to evaluate the physician’s characteristics such as age, years of practice, or specialist training that could influence the counseling of the patient regarding the choice of conservative or radical treatment. However, surveys conducted among gynecologists have found that patient age, surgical approach, the specialty and gender of the doctor mostly influence the decision of bilateral salpingo-oophorectomy [49].

Few studies have evaluated the patient’s preference for bilateral salpingo-oophorectomy or conservative method; a study has shown that some patients are influenced by the risk of developing ovarian cancer or the need for hormone replacement therapy, while another study found that personality traits, sexual activity and preoperative symptoms affect all decisions [49]. Age, low level of education, lack of knowledge and understanding, social status—multiple previous abdominal surgeries are important factors that affect decision making when patients choose to conserve ovaries and prophylactic oophorectomy during hysterectomy for benign lesions. In addition, women also depend on the gynecologist to make an appropriate decision. Therefore, we need to reconsider the age at which we recommend prophylactic oophorectomy. We must highlight the benefits of conserving the ovaries and the side effects of prophylactic oophorectomy in low-risk women. Negative counseling should be avoided [50].

According to a study by Catherine A. Matthews, regarding the management strategies for ovaries at the time of hysterectomy for benign disease, every woman undergoing hysterectomy for benign disease is faced with the complex and controversial decision of what to do with ancillary uterine structures, the fallopian tubes and ovaries. Many women feel inadequately informed about their treatment options. Gynecologic surgeons will likely play the most influential role in making woman’s medical decisions. Prophylactic BSO should be performed in high risk patients, with a personal history of breast cancer or grade 1 and 2 relatives with breast or ovarian cancer. For patients with low risk, with no evidence of a hereditary ovarian or breast cancer, ovarian preservation should be performed for premenopausal women. For postmenopausal women, no conclusive evidence for ovarian conservation exists for coronary heart disease prevention, hip fracture prevention, or improved sexual function. Leaving the ovaries in situ increased the risk of incidence for breast and ovarian cancer and the need for subsequent adnexal surgery for benign and malignant disease. Elective BSO should be performed on patients with increased surgical risk, obesity and for patients with low surgical risk the medical decision should be common. Opportunistic salpingectomy should be universally encouraged [51].

Sarah Miles MD, an ovary conservationist and researcher, said she did not know a period when oophorectomy was not standard and commented on her research on estrogen deficiency. She has stated that there is a difference with and without estrogen deficiency, with cognitive decline, memory loss, dementia and loss of bone density. Coronary vascular disease was the worst thing to shorten the life of people. An average of 300 oophorectomies should be performed in order to prevent one death from ovarian cancer. She said that ‘‘Leaving them inside is a smarter medicine” [52].

## 9. Conclusions

The decision for elective oophorectomy should be made by correct evaluation of the cost-benefit ratio, the advantages/disadvantages, the immediate and long-term risks and complications.

Reducing the indication of surgical treatment for asymptomatic fibroma, conserving the ovaries in low risk patients, postponed or conservative treatment of ovarian cysts that are part of the IOTA benignity criteria could reduce surgical procedures.

Multidisciplinary pre-operative counseling to understand the risks of early surgically induced menopause (which can be supplemented by appropriate gynecologist’s knowledge of hormonal and metabolic consequences) is necessary for the patient to become the primary decision maker.

## Figures and Tables

**Table 1 medicina-55-00482-t001:** Bilateral salpingectomy with ovarian retention (BSOR).

The Advantages of Salpingectomy	The Disadvantages of Salpingectomy
1. Removing a proportion of the risk of women with BRCA 1/2 mutations that refuse BSO before menopause. Postponing a premature menopause	1. The incidence rate of tubal origin of ovarian cancer is not known accurately.
2. Postponing a premature menopause	2. The mechanism by which tubal ligation mediates risk reduction is not known; the benefit is strictly related to the reduction of the risk of tubal cancer or also affects the risk of primary ovarian / peritoneal cancer.
3. The possibility of performing a laparoscopic procedure almost in all cases; the opportunity of peritoneal cavity inspection and peritoneal biopsy.	3. Personally postponing BO after BSOP for a long time, increasing the cumulative risk of ovarian cancer.
4. Provides the possibility of fertility preservation through assisted techniques.	4. The difficulty of making the right decision for the use of BSOP due to the risk differences between BRCA 1 and BRCA 2 in women aged 30–50 years; with a mean age at diagnosis 51,2 years for BRCA 1 and 57,2 for BRCA 2; there is a tendency for delayed BSOR for BRCA 2.
5. Easier procedure to accept for women who do not want BOS for psychological reasons.	5. Reducing the risk of breast cancer by postponing BO; the maximum reduction in breast cancer risk was observed when performing BO before natural menopause
	6. Increasing the risk of surgical complications at BO because it becomes a reintervention.

**Table 2 medicina-55-00482-t002:** The effects of surgically induced menopause.

The Effects of Surgically Induced Menopause	
global survival	- the mortality risk increases in patients with bilateral oophorectomy under 45–50 years of age without hormone replacement therapy (HRT) - conservation of ovaries in women under 65 years with low risk for ovarian cancer provides long-term survival benefit - decrease by 2% of total mortality with each year of menopause delay - ovarian conservation is associated with lower death rates regardless of the cause
cardiovascular disease	- 40% increased risk for cardiovascular disease in women with oophorectomy without hormone replacement therapy (HRT) regardless of age - the highest risk for cardiovascular diseases in women <45–50 years without HRT or interrupted HRT - proposed mechanism: accelerated atherosclerosis due to increased levels of atherogenic lipoproteins secondary to a hypoestrogenic state - HRT eliminates the risk
cognitive function	- increased risk of cognitive impairment, dementia or Parkinsonism and Alzheimer’s disease - younger age at the time of surgically induced menopause is associated with a rapid decline in global cognitive function - use of HRT for at least 10 years, with initiation in the first 5 years after menopause onset, is associated with improvement in cognitive decline
sexual function	- negative consequences on sexual function in women with oophorectomy without HRT, such as: decreased libido and difficult sexual arousal, a 3-fold risk of anorgasmia; increased risk of hypoactive sexual desire disorder
bone loss	- bilateral oophorectomy in women under 45 years of age is considered a risk factor for osteoporosis - bilateral oophorectomy increases the risk of osteoporosis even in postmenopausal women compared to those with intact ovaries

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
