# Peer review of "Surgically Induced Menopause—A Practical Review of Literature"

_1010-660X, 2019, doi:10.3390/medicina55080482_

Round 1

Reviewer 1 Report

This is a practically useful review of the literature devoted to an important clinical problem. However: 

The text needs extensive language editing as it contains errors and improper terminology (for example "premenopausal" and not "non menopausal"). 

The abbreviations used for the 1st time in the text should be proceeded with full names (for example ESMO-ESGO-ESTRO, ACOG etc.). 

The Authors finish their manuscript with chapter "Instead of the conclusions" which I think is an improper way to summarise the content of such review paper - there always should a conclusion even if it is like  "the available data do not provide the univocal answer to the question ....".

Author Response

Thank you very much for your appreciation.

According to your recommendations I have edited the grammar and terminology errors. The abbreviations were written in full name. Also I have replaced the final paragraph title with "Conclusion".

Reviewer 2 Report

This is a very important topic and its nice to see a review done. The paper requires professional English editing before it is acceptable for publication.

It was difficult to figure out how the manuscript was set up and how to follow along. Please add a summary overview of the manuscript which could be included in the introduction so the reader knows how to follow along with the paper. Also, there is little tying each section together (it reads as though someone individually summarized each block, without finding a cohesive way to put the review together). It should be a cohesive message throughout.

My specific comments are below:

- We use the term oophorectomy clinically in the US to discuss surgical removal of both ovaries (not ovariectomy).

- The term 'climacterium' is not commonly used any longer and not necessary in the introduction.

- You are missing the other part of your parentheses in the intro after 'decreased elasticity.'

- You do not need 'the' in front of  sexual function.

- Please include references to support your statements about menopausal changes and symptoms.

- I would delete "and debate" before over the years in the last sentence of the introduction. It is well established there are many differences between natural and surgical induced menopause.

- It is unclear if the ovary continues to produce DEHA and androstenedione after menopause, and more recent studies suggest those levels may be coming from the andrenal glands. (Couzinet      B, Meduri G, Lecce MG, Young J, Brailly S, Loosfelt H, Milgrow E, Schaison      G. The Postmenopausal Ovary is Not a Major Androgen-Producing Gland. The Journal of Clinical Endocrinology & Metabolism, Volume 86, Issue      10, 1 October 2001, Pages 5060–5066, https://doi.org/10.1210/jcem.86.10.7900). Please update for accuracy.

- line 37 add been to 'has generally faster'

- important papers are missing from your references for lines 38 - 41 (include work be Dr. Walter Rocca here).

- Section 4 starting on line 66 is highlighted different than the other sections? Why is that? There are other similar sections throughout. Please edit or clarify.

- I would put as much of the pro vs. con information in Tables and if its in tables it does not need to be written out extensively in the text, just refer to the tables (such as Table 1)

- The section headings that start with 'on' need to be capitalized.

- Could you put the sections summarizing the effects of surgically induced menopause into a large table?

- I don't not understand the ending? 'instead of the conclusions'??? And the first word of the conclusions is not capitalized. Your conclusion appears to be a rephrasing of Dr. Faubion's paper, as opposed to your conclusion and recommendations.

Author Response

Thank you very much for your appreciation. 

New paragraphs were added, to give the manuscript a logical and cohesive course and to tie the section together. The requested summary overview was added in the introduction. Also, a professional English editing was performed. The manuscript, as you will see, was heavily edited (using Track Changes) trying to be as thoroughly as possible.

We followed every comment and made changes accordingly.

1. We use the term oophorectomy clinically in the US to discuss surgical removal of both ovaries (not ovariectomy).

Response 1: the term ovariectomy was replaced with oophorectomy.

2. The term 'climacterium' is not commonly used any longer and not necessary in the introduction.

Response 2: the term climacterium was removed.

3. You are missing the other part of your parentheses in the intro after 'decreased elasticity.'

Response 3: the other parentheses was added.

4. You do not need 'the' in front of sexual function.

Response 4: “the” was removed.

5. Please include references to support your statements about menopausal changes and symptoms.

Response 5: we included 2 new references to support those statements

6. I would delete "and debate" before over the years in the last sentence of the introduction. It is well established there are many differences between natural and surgical induced menopause.

Response 6: the entire sentence was removed.

7. It is unclear if the ovary continues to produce DEHA and androstenedione after menopause, and more recent studies suggest those levels may be coming from the andrenal glands.

Response 7: The initial text was replaced with your suggested study. (Couzinet      B, Meduri G, Lecce MG, Young J, Brailly S, Loosfelt H, Milgrow E, Schaison      G. The Postmenopausal Ovary is Not a Major Androgen-Producing Gland. The      Journal of Clinical Endocrinology & Metabolism, Volume 86, Issue      10, 1 October 2001, Pages 5060–5066, https://doi.org/10.1210/jcem.86.10.7900).

8. Line 37 add been to 'has generally faster'.

Response 8: We removed these words entirely.

9. Important papers are missing from your references for lines 38 - 41 (include work be Dr. Walter Rocca here).

Response 9: Rocca’s studies have been cited.

10. Section 4 starting on line 66 is highlighted different than the other sections? Why is that? There are other similar sections throughout. Please edit or clarify.

Response 10: the highlight has been removed.

11. I would put as much of the pro vs. con information in Tables and if its in tables it does not need to be written out extensively in the text, just refer to the tables (such as Table 1)

Response 11: We appreciate de suggestion, but we consider more appropriate and comprehensive if the information is presented as a text.

12. The section headings that start with 'on' need to be capitalized.

Response 12: We capitalized all section starting with “On”.

13. Could you put the sections summarizing the effects of surgically induced menopause into a large table?

Response 13: We included the suggested table.

14. I don't not understand the ending? 'instead of the conclusions'??? And the first word of the conclusions is not capitalized. Your conclusion appears to be a rephrasing of Dr. Faubion's paper, as opposed to your conclusion and recommendations.

Response 14: We modified the conclusions and we remove the part from Faubion’s paper. Our opinion regarding the subject of this review is presented in the conclusion.